# Decision Jungles:
# Compact and Rich Models for Classification

**Jamie Shotton**    **Toby Sharp**    **Pushmeet Kohli**
**Sebastian Nowozin**    **John Winn**    **Antonio Criminisi**
Microsoft Research

## Abstract

Randomized decision trees and forests have a rich history in machine learning and have seen considerable success in application, perhaps particularly so for computer vision. However, they face a fundamental limitation: given enough data, the number of nodes in decision trees will grow exponentially with depth. For certain applications, for example on mobile or embedded processors, memory is a limited resource, and so the exponential growth of trees limits their depth, and thus their potential accuracy. This paper proposes *decision jungles*, revisiting the idea of ensembles of rooted decision directed acyclic graphs (DAGs), and shows these to be compact and powerful discriminative models for classification. Unlike conventional decision trees that only allow one path to every node, a DAG in a decision jungle allows multiple paths from the root to each leaf. We present and compare two new node merging algorithms that jointly optimize both the features and the structure of the DAGs efficiently. During training, node splitting and node merging are driven by the minimization of exactly the same objective function, here the weighted sum of entropies at the leaves. Results on varied datasets show that, compared to decision forests and several other baselines, decision jungles require dramatically less memory while considerably improving generalization.

## 1 Introduction

Decision trees have a long history in machine learning and were one of the first models proposed for inductive learning [14]. Their use for classification and regression was popularized by the work of Breiman [6]. More recently, they have become popular in fields such as computer vision and information retrieval, partly due to their ability to handle large amounts of data and make efficient predictions. This has led to successes in tasks such as human pose estimation in depth images [29].

Although trees allow making predictions efficiently, learning the optimal decision tree is an NP-hard problem [15]. In his seminal work, Quinlan proposed efficient approximate methods for learning decision trees [27, 28]. Some researchers have argued that learning optimal decision trees could be harmful as it may lead to overfitting [21]. Overfitting may be reduced by controlling the model complexity, *e.g.* via various stopping criteria such as limiting the tree depth, and post-hoc pruning.

These techniques for controlling model complexity impose implicit limits on the type of classification boundaries and feature partitions that can be induced by the decision tree. A number of approaches have been proposed in the literature to regularize tree models without limiting their modelling power. The work in [7] introduced a non-greedy Bayesian sampling-based approach for constructing decision trees. A prior over the space of trees and their parameters induces a posterior distribution, which can be used, for example, to marginalize over all tree models. There are similarities between the idea of randomly drawing multiple trees via a Bayesian procedure and construction of random tree ensembles (forests) using bagging, a method shown to be effective in many applications [1, 5, 9]. Another approach to improve generalization is via large-margin tree classifiers [4].

While the above-mentioned methods can reduce overfitting, decision trees face a fundamental limitation: their exponential growth with depth. For large datasets where deep trees have been shown to be more accurate than large ensembles (*e.g.* [29]), this exponential growth poses a problem for implementing tree models on memory-constrained hardware such as embedded or mobile processors.

In this paper, we investigate the use of randomized ensembles of rooted decision directed acyclic graphs (DAGs) as a means to obtain compact and yet accurate classifiers. We call these ensembles 'decision jungles', after the popular 'decision forests'. We formulate the task of learning each DAG in a jungle as an energy minimization problem. Building on the information gain measure commonly used for training decision trees, we propose an objective that is defined jointly over the features of the split nodes *and* the structure of the DAG. We then propose two minimization methods for learning the optimal DAG. Both methods alternate between optimizing the split functions at the nodes of the DAG and optimizing the placement of the branches emanating from the parent nodes. As detailed later, they differ in the way they optimize the placement of branches.

We evaluate jungles on a number of challenging labelling problems. Our experiments below quantify a substantially reduced memory footprint for decision jungles compared to standard decision forests and several baselines. Furthermore, the experiments also show an important side-benefit of jungles: our optimization strategy is able to achieve considerably improved generalization for only a small extra cost in the number of features evaluated per test example.

**Background and Prior Work.** The use of rooted decision DAGs ('DAGs' for short) has been explored by a number of papers in the literature. In [16, 26], DAGs were used to combine the outputs of $C \times C$ binary 1-v-1 SVM classifiers into a single $C$-class classifier. More recently, in [3], DAGs were shown to be a generalization of cascaded boosting.

It has also been shown that DAGs lead to accurate predictions while having lower model complexity, subtree replication, and training data fragmentation compared to decision trees. Most existing algorithms for learning DAGs involve training a conventional tree that is later manipulated into a DAG. For instance [17] merges same-level nodes which are associated with the same split function. They report performance similar to that of C4.5-trained trees, but with a much reduced number of nodes. Oliveira [23] used local search method for constructing DAGs in which tree nodes are removed or merged together based on similarity of the underlying sub-graphs and the corresponding message length reduction. A message-length criterion is also employed by the node merging algorithm in [24]. Chou [8] investigated a $k$-means clustering for learning decision trees and DAGs (similar 'ClusterSearch' below), though did not jointly optimize the features with the DAG structure. Most existing work on DAGs have focused on showing how the size and complexity of the learned tree model can be reduced without substantially degrading its accuracy. However, their use for *increasing* test accuracy has attracted comparatively little attention [10, 20, 23].

In this paper we show how jungles, ensembles of DAGs, optimized so as to reduce a well defined objective function, can produce results which are superior to those of analogous decision tree ensembles, both in terms of model compactness as well as generalization. Our work is related to [25], where the authors achieve compact classification DAGs via post-training removal of redundant sub-trees in forests. In contrast, our probabilistic node merging is applied directly and efficiently during training, and both saves space as well as achieves greater generalization for multi-class classification.

**Contributions.** In summary, our contributions are: (i) we highlight that traditional decision trees grow exponentially in memory with depth, and propose decision jungles as a means to avoid this; (ii) we propose and compare two learning algorithms that, within each level, jointly optimize an objective function over both the structure of the graph and the features; (iii) we show that not only do the jungles dramatically reduce memory consumption, but can also improve generalization.

## 2    Forests and Jungles

Before delving into the details of our method for learning decision jungles, we first briefly discuss how decision trees and forests are used for classification problems and how they relate to jungles.

**Binary decision trees.** A binary decision tree is composed of a set of nodes each with an in-degree of 1, except the root node. The out-degree for every internal (split) node of the tree is 2 and for the leaf nodes is 0. Each split node contains a binary split function ('feature') which decides whether an

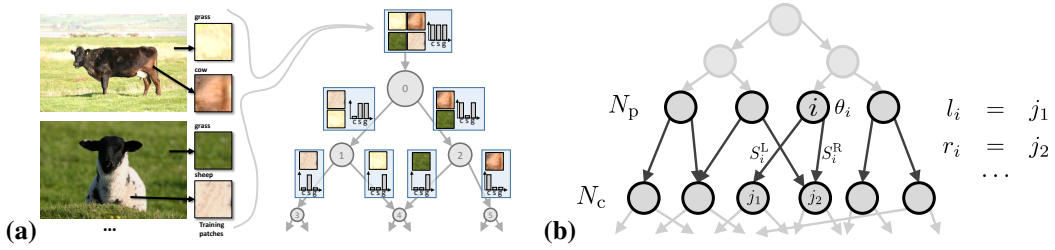

Figure 1: **Motivation and notation.** **(a)** An example use of a rooted decision DAG for classifying image patches as belonging to grass, cow or sheep classes. Using DAGs instead of trees reduces the number of nodes and can result in better generalization. For example, differently coloured patches of grass (yellow and green) are merged together into node 4, because of similar class statistics. This may encourage generalization by representing the fact that grass may appear as a mix of yellow and green. **(b)** Notation for a DAG, its nodes, features and branches. See text for details.

input instance that reaches that node should progress through the left or right branch emanating from the node. Prediction in binary decision trees involves every input starting at the root and moving down as dictated by the split functions encountered at the split nodes. Prediction concludes when the instance reaches a leaf node, each of which contains a unique prediction. For classification trees, this prediction is a normalized histogram over class labels.

**Rooted binary decision DAGs.** Rooted binary DAGs have a different architecture compared to decision trees and were introduced by Platt *et al.* [26] as a way of combining binary classifier for multi-class classification tasks. More specifically a rooted binary DAG has: (i) one root node, with in-degree 0; (ii) multiple split nodes, with in-degree $\geq 1$ and out-degree 2; (iii) multiple leaf nodes, with in-degree $\geq 1$ and out-degree 0. Note that in contrast to [26], if we have a $C$-class classification problem, here we do not necessarily expect to have $C$ DAG leaves. In fact, the leaf nodes are not necessarily pure; And each leaf remains associated with an empirical class distribution.

**Classification DAGs vs classification trees.** We explain the relationship between decision trees and decision DAGs using the image classification task illustrated in Fig. 1(a) as an example. We wish to classify image patches into the classes: cow, sheep or grass. A labelled set of patches is used to train a DAG. Since patches corresponding to different classes may have different average intensity, the root node may decide to split them according to this feature. Similarly, the two child nodes may decide to split the patches further based on their chromaticity. This results in grass patches with different intensity and chromaticity (bright yellow and dark green) ending up in different subtrees. However, if we detect that two such nodes are associated with similar class distributions (peaked around grass in this case) and merge them, then we get a single node with training examples from both grass types. This helps capture the degree of variability intrinsic to the training data, *and* reduce the classifier complexity. While this is clearly a toy example, we hope it gives some intuition as to why rooted DAGs are expected to achieve the improved generalization demonstrated in Section 4.

## 3 Learning Decision Jungles

We train each rooted decision DAG in a jungle independently, though there is scope for merging across DAGs as future work. Our method for training DAGs works by growing the DAG one level at a time.[1] At each level, the algorithm jointly learns the features and branching structure of the nodes. This is done by minimizing an objective function defined over the predictions made by the child nodes emanating from the nodes whose split features are being learned.

Consider the set of nodes at two consecutive levels of the decision DAG (as shown in Fig. 1b). This set consist of the set of parent nodes $N_\mathrm{p}$ and a set of child nodes $N_\mathrm{c}$. We assume in this work a known value for $M = |N_\mathrm{c}|$. $M$ is a parameter of our method and may vary per level. Let $\theta_i$ denote the parameters of the split feature function $f$ for parent node $i \in N_\mathrm{p}$, and $S_i$ denote the set of labelled training instances $(x, y)$ that reach node $i$. Given $\theta_i$ and $S_i$, we can compute the set of instances from node $i$ that travel through its left and right branches as $S_i^\mathrm{L}(\theta_i) = \{(x, y) \in S_i \mid f(\theta_i, x) \leq 0\}$

and $S_i^{\mathrm{R}}(\theta_i) = S_i \setminus S_i^{\mathrm{L}}(\theta_i)$, respectively. We use $l_i \in N_{\mathrm{c}}$ to denote the current assignment of the left outwards edge from parent node $i \in N_{\mathrm{p}}$ to a child node, and similarly $r_i \in N_{\mathrm{c}}$ for the right outward edge. Then, the set of instances that reach any child node $j \in N_{\mathrm{c}}$ is:

$$S_j(\{\theta_i\}, \{l_i\}, \{r_i\}) = \left[ \bigcup_{i \in N_{\mathrm{p}} \text{ s.t. } l_i = j} S_i^{\mathrm{L}}(\theta_i) \right] \cup \left[ \bigcup_{i \in N_{\mathrm{p}} \text{ s.t. } r_i = j} S_i^{\mathrm{R}}(\theta_i) \right]. \tag{1}$$

The objective function $E$ associated with the current level of the DAG is a function of $\{S_j\}_{j \in N_{\mathrm{c}}}$. We can now formulate the problem of learning the parameters of the decision DAG as a joint minimization of the objective over the split parameters $\{\theta_i\}$ *and* the child assignments $\{l_i\}, \{r_i\}$. Thus, the task of learning the current level of a DAG can be written as:

$$\min_{\{\theta_i\}, \{l_i\}, \{r_i\}} E(\{\theta_i\}, \{l_i\}, \{r_i\}). \tag{2}$$

**Maximizing the Information Gain.** Although our method can be used for optimizing any objective $E$ that decomposes over nodes, including in theory a regression-based objective, for the sake of simplicity we focus in this work on the information gain objective commonly used for classification problems. The information gain objective requires the minimization of the total weighted entropy of instances, defined as:

$$E(\{\theta_i\}, \{l_i\}, \{r_i\}) = \sum_{j \in N_{\mathrm{c}}} |S_j| \, H(S_j) \tag{3}$$

where $S_j$ is defined in (1), and $H(S)$ is the Shannon entropy of the class labels $y$ in the training instances $(x, y) \in S$.

Note that if the number of child nodes $M$ is equal to twice the number of parent nodes *i.e.* $M = 2|N_p|$, then the DAG becomes a tree and we can optimize the parameters of the different nodes independently, as done in standard decision tree training, to achieve optimal results.

## 3.1 Optimization

The minimization problem described in (2) is hard to solve exactly. We propose two local search based algorithms for its solution: LSearch and ClusterSearch. As local optimizations, neither are likely to reach a global minimum, but in practice both are effective at minimizing the objective. The experiments below show that the simpler LSearch appears to be more effective.

**LSearch.** The LSearch method starts from a feasible assignment of the parameters, and then alternates between two coordinate descent steps. In the first (split-optimization) step, it sequentially goes over every parent node $k$ in turn and tries to find the split function parameters $\theta_k$ that minimize the objective function, keeping the values of $\{l_i\}, \{r_i\}$ and the split parameters of all other nodes fixed:

$$\text{for } k \in N_{\mathrm{p}}$$
$$\theta_k \leftarrow \operatorname*{argmin}_{\theta_k'} E(\theta_k' \cup \{\theta_i\}_{i \in N_{\mathrm{p}} \setminus \{k\}}, \{l_i\}, \{r_i\})$$

This minimization over $\theta_k'$ is done by random sampling in a manner similar to decision forest training [9]. In the second (branch-optimization) step, the algorithm redirects the branches emanating from each parent node to different child nodes, so as to yield a lower objective:

$$\text{for } k \in N_{\mathrm{p}}$$
$$l_k \leftarrow \operatorname*{argmin}_{l_k' \in N_{\mathrm{c}}} E(\{\theta_i\}, l_k' \cup \{l_i\}_{i \in N_{\mathrm{p}} \setminus \{k\}}, \{r_i\})$$
$$r_k \leftarrow \operatorname*{argmin}_{r_k' \in N_{\mathrm{c}}} E(\{\theta_i\}, \{l_i\}, r_k' \cup \{r_i\}_{i \in N_{\mathrm{p}} \setminus \{k\}})$$

The algorithm terminates when no changes are made, and is guaranteed to converge. We found that a greedy initialization of LSearch (allocating splits to the most energetic parent nodes first) resulted in a lower objective after optimization than a random initialization. We also found that a stochastic version of the above algorithm where only a single randomly chosen node was optimized at a time resulted in similar reductions in the objective for considerably less compute.

**ClusterSearch.** The ClusterSearch algorithm also alternates between optimizing the branching variables and the split parameters, but differs in that it optimizes the branching variables more globally. First, $2|N_{\mathrm{p}}|$ temporary child nodes are built via conventional tree-based, training-objective minimization procedures. Second, the temporary nodes are clustered into $M = |N_{\mathrm{c}}|$ groups to produce a DAG. Node clustering is done via the Bregman information objective optimization technique in [2].

## 4 Experiments and results

This section compares testing accuracy and computational performance of our decision jungles with state-of-the-art forests of binary decision trees and their variants on several classification problems.

### 4.1 Classification Tasks and Datasets

We focus on semantic image segmentation (pixel-wise classification) tasks, where decision forests have proven very successful [9, 19, 29]. We evaluate our jungle model on the following datasets:

**(A) Kinect body part classification** [29] (31 classes). We train each tree or DAG in the ensemble on a separate 1000 training images with 250 example pixels randomly sampled per image. Following [29], 3 trees or DAGs are used unless otherwise specified. We test on (a common set of) 1000 images drawn randomly from the MSRC-5000 test set [29]. We use a DAG merging schedule of $|N_{\mathrm{c}}^{D}| = \min(M, 2^{\min(5,D)} \cdot 1.2^{\max(0,D-5)})$, where $M$ is a fixed constant maximum width and $D$ is the current level (depth) in the tree.

**(B) Facial features segmentation** [18] (8 classes including background). We train each of 3 trees or DAGs in the ensemble on a separate 1000 training images using every pixel. We use a DAG merging schedule of $|N_{\mathrm{c}}^{D}| = \min(M, 2^{D})$.

**(C) Stanford background dataset** [12] (8 classes). We train on all 715 labelled images, seeding our feature generator differently for each of 3 trees or DAGs in the ensemble. Again, we use a DAG merging schedule of $|N_{\mathrm{c}}^{D}| = \min(M, 2^{D})$.

**(D) UCI data sets** [22]. We use 28 classification data sets from the UCI corpus as prepared on the libsvm data set repository.[2] For each data set all instances from the training, validation, and test set, if available, are combined to a large set of instances. We repeat the following procedure five times: randomly permute the instances, and divide them 50/50 into training and testing set. Train on the training set, evaluate the multiclass accuracy on the test set. We use 8 trees or DAGs per ensemble. Further details regarding parameter choices can be found in the supplementary material.

For all segmentation tasks we use the Jaccard index (intersection over union) as adopted in PASCAL VOC [11]. Note that this measure is stricter than *e.g.* the per class average metric reported in [29]. On the UCI dataset we report the standard classification accuracy numbers. In order to keep training time low, the training sets are somewhat reduced compared to the original sources, especially for (A). However, identical trends were observed in limited experiments with more training data.

### 4.2 Baseline Algorithms

We compare our decision jungles with several tree-based alternatives, listed below.

**Standard Forests of Trees.** We have implemented standard classification forests, as described in [9] and building upon their publically available implementation.

**Baseline 1: Fixed-Width Trees (A).** As a first variant on forests, we train binary decision trees with an enforced maximum width $M$ at each level, and thus a reduced memory footprint. This is useful to tease out whether the improved generalization of jungles is due more to the reduced model complexity or to the node merging. Training a tree with fixed width is achieved by ranking the leaf nodes $i$ at each level by decreasing value of $E(S_i)$ and then greedily splitting only the $M/2$ nodes with highest value of the objective. The leaves that are not split are discarded.

**Baseline 2: Fixed-Width Trees (B).** A related, second tree-based variant is obtained by greedily optimizing the best split candidate for all leaf nodes, then ranking the leaves by reduction in the

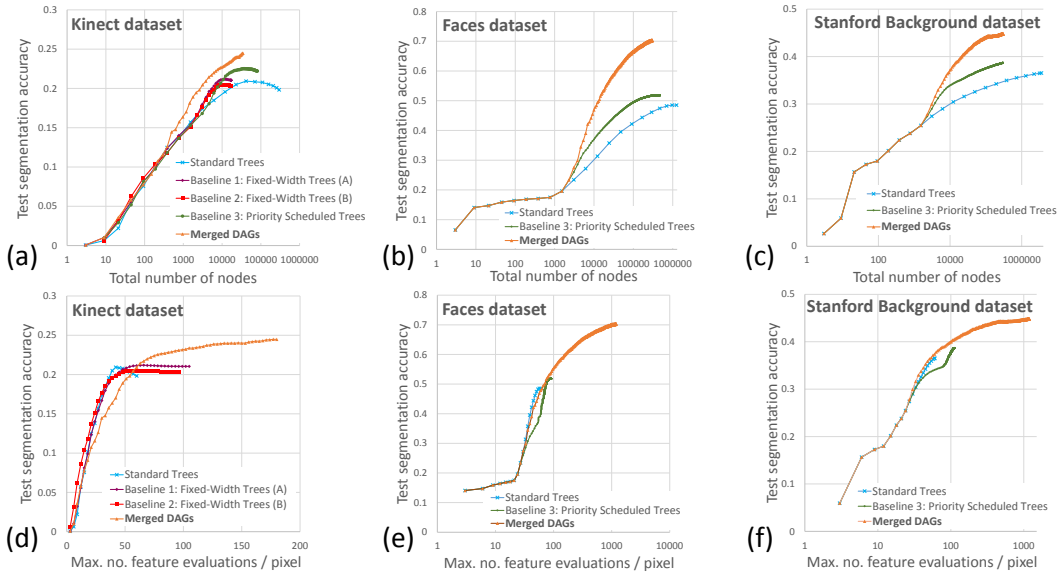

Figure 2: **Accuracy comparisons.** Each graph compares Jaccard scores for jungles *vs.* standard decision forests and three other baselines. **(a, b, c)** Segmentation accuracy as a function of the total number of nodes in the ensemble (*i.e.* memory usage) for three different datasets. **(d, e, f)** Segmentation accuracy as a function of the maximum number of test comparisons per pixel (maximum depth $\times$ size of ensemble), for the same datasets. Jungles achieve the same accuracy with fewer nodes. Jungles also improve the overall generalization of the resulting classifier.

objective, and greedily taking only the $M/2$ splits that most reduce the objective.[3] The leaf nodes that are not split are discarded from further consideration.

**Baseline 3: Priority Scheduled Trees.** As a final variant, we consider priority-driven tree training. Current leaf nodes are ranked by the reduction in the objective that would be achieved by splitting them. At each iteration, the top $M$ nodes are split, optimal splits computed and the new children added into the priority queue. This baseline is identical to the baseline 2 above, except that nodes that are not split at a particular iteration are part of the ranking at subsequent iterations. This can be seen as a form of tree pruning [13], and in the limit, will result in standard binary decision trees. As shown later, the trees at intermediate iterations can give surprisingly good generalization.

### 4.3 Comparative Experiments

**Prediction Accuracy *vs.* Model Size.** One of our two main hypotheses is that jungles can reduce the amount of memory used compared to forests. To investigate this we compared jungles to the baseline forests on three different datasets. The results are shown in Fig. 2 (top row). Note that the jungles of merged DAGs achieve the same accuracy as the baselines with substantially fewer total nodes. For example, on the Kinect dataset, to achieve an accuracy of 0.2, the jungle requires around 3000 nodes whereas the standard forest require around 22000 nodes. We use the total number of nodes as a proxy for memory usage; the two are strongly linked, and the proxy works well in practice. For example, the forest of 3 trees occupied 80MB on the Kinect dataset *vs.* 9MB for a jungle of 3 DAGs. On the Faces dataset the forest of 3 trees occupied 7.17MB *vs.* 1.72MB for 3 DAGs.

A second hypothesis is that merging provides a good way to regularize the training and thus increases generalization. Firstly, observe how all tree-based baselines saturate and in some cases start to overfit as the trees become larger. This is a common effect with deep trees and small ensembles. Our merged DAGs appear to be able to avoid this overfitting (at least in as far as we have trained them here), and further, actually have increased the generalization quite considerably.

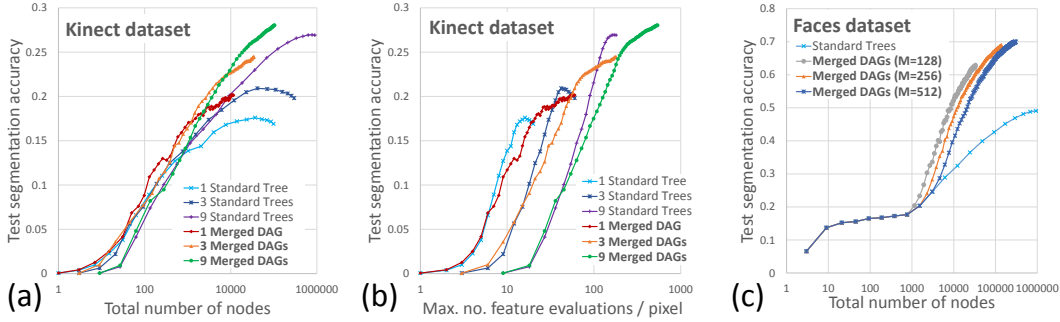

Figure 3: **(a, b) Effect of ensemble size on test accuracy.** (a) plots accuracy against the total number of nodes in the ensemble, whereas (b) plots accuracy against the maximum number of computations required at test time. For a fixed ensemble size jungles of DAGs achieve consistently better generalization than conventional forests. **(c) Effect of merging parameter $M$ on test accuracy.** The model width $M$ has a regularizing effect on our DAG model. For other results shown on this dataset, we set $M = 256$. See text for details.

Interestingly, the width-limited tree-based baselines perform substantially better than the standard tree training algorithm, and in particular the priority scheduling appears to work very well, though still inferior to our DAG model. This suggests that both reducing the model size *and* node merging have a substantial positive effect on generalization.

**Prediction Accuracy *vs*. Depth.** We do not expect the reduction in memory given by merging to come for free: there is likely to be a cost in terms of the number of nodes evaluated for any individual test example. Fig. 2 (bottom row) shows this trade-off. The large gains in memory footprint and accuracy come at a relatively small cost in the number of feature evaluations at test time. Again, however, the improved generalization is also evident. The need to train deeper also has some effect on training time. For example, training 3 trees for Kinect took 32mins *vs*. 50mins for 3 DAGs.

**Effect of Ensemble Size.** Fig. 3 (a, b) compares results for 1, 3, and 9 trees/DAGs in a forest/jungle. Note from (a) that in all cases, a jungle of DAGs uses substantially less memory than a standard forest for the same accuracy, and also that the merging consistently increases generalization. In (b) we can see again that this comes at a cost in terms of test time evaluations, but note that the upper-envelope of the curves belongs in several regions to DAGs rather than trees.

**LSearch *vs*. ClusterSearch Optimization.** In experiments we observed the LSearch algorithm to perform better than the ClusterSearch optimization, both in terms of the objective achieved (reported in the table below for the face dataset) and also in test accuracy. The difference is slight, yet very consistent. In our experiments the LSearch algorithm was used with 250 iterations.

| Number of nodes | 2047 | 5631 | 10239 | 20223 | 30207 | 40191 |
|---|---|---|---|---|---|---|
| LSearch objective | 0.735 | 0.596 | 0.514 | 0.423 | 0.375 | 0.343 |
| ClusterSearch objective | 0.739 | 0.605 | 0.524 | 0.432 | 0.382 | 0.351 |

**Effect of Model Width.** We performed an experiment investigating changes to $M$, the maximum tree width. Fig. 3 (c) shows the results. The merged DAGs consistently outperform the standard trees both in terms of memory consumption and generalization, for all settings of $M$ evaluated. Smaller values of $M$ improve accuracy while keeping memory constant, but must be trained deeper.

**Qualitative Image Segmentation Results.** Fig. 4 shows some randomly chosen segmentation results on both the Kinect and Faces data. On the Kinect data, forests of 9 trees are compared to jungles of 9 DAGs. The jungles appear to give smoother segmentations than the standard forests, perhaps more so than the quantitative results would suggest. On the Faces data, small forests of 3 trees are compared to jungles of 3 DAGs, with each model containing only 48k nodes in total.

**Results on UCI Datasets.** Figure 5 reports the test classification accuracy as a function of model size for two UCI data sets. The full results for all UCI data sets are reported in the supplementary material. Overall using DAGs allows us to achieve higher accuracies at smaller model sizes, but in

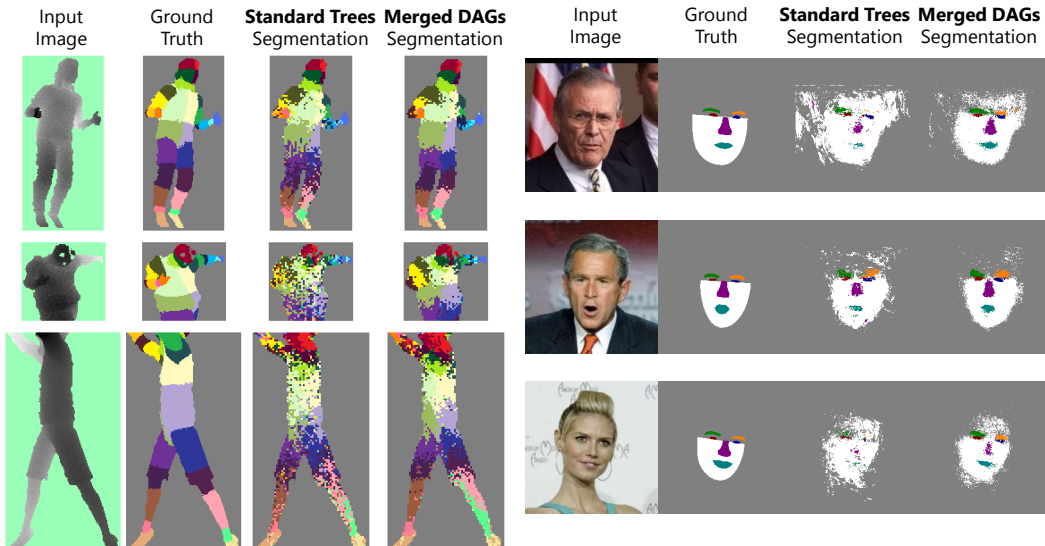

Figure 4: **Qualitative results.** A few example results on the Kinect body parts and face segmentation tasks, comparing standard trees and merged DAGs with the same number of nodes.

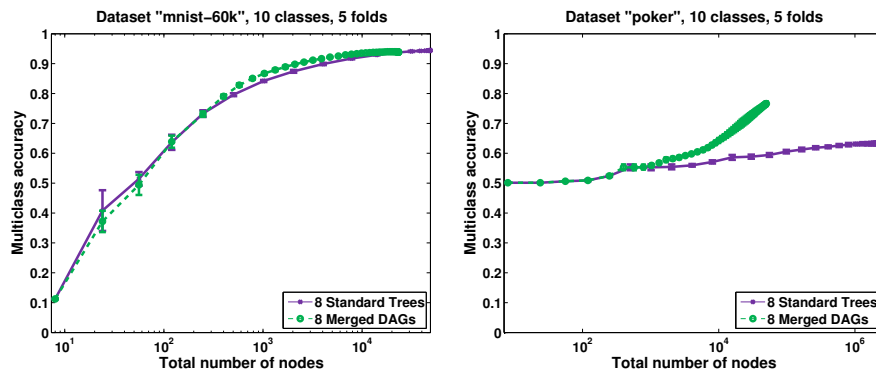

Figure 5: **UCI classification results** for two data sets, MNIST-60k and Poker, eight trees or DAGs per ensemble. The MNIST result is typical in that the accuracy improvements of DAGs over trees is small but achieved at a smaller number of nodes (memory). The largest UCI data set (Poker, 1M instances) profits most from the use of randomized DAGs.

most cases the generalization performance is not improved or only slightly improved. The largest improvements for DAGs over trees is reported for the largest dataset (Poker).

## 5 Conclusion

This paper has presented decision jungles as ensembles of rooted decision DAGs. These DAGs are trained, level-by-level, by jointly optimizing an objective function over both the choice of split function and the structure of the DAG. Two local optimization strategies were evaluated, with an efficient move-making algorithm producing the best results. Our evaluation on a number of diverse and challenging classification tasks has shown jungles to improve both memory efficiency and generalization for several tasks compared to conventional decision forests and their variants.

We believe that decision jungles can be extended to regression tasks. We also plan to investigate multiply rooted trees and merging between DAGs within a jungle.

**Acknowledgements.** The authors would like to thank Albert Montillo for initial investigation of related ideas.

## Footnotes

[1]Jointly training all levels of the tree simultaneously remains an expensive operation [15].

[2] http://www.csie.ntu.edu.tw/~cjlin/libsvmtools/datasets/

[3]In other words, baseline 1 optimizes the most energetic nodes, whereas baseline 2 optimizes all nodes and takes only the splits that most reduce the objective.

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
