[Supplementary Material]

# Decision Jungles:
# Compact and Rich Models for Classification
# *Supplementary Material*

**Jamie Shotton**      **Toby Sharp**      **Pushmeet Kohli**
**Sebastian Nowozin**      **John Winn**      **Antonio Criminisi**
Microsoft Research, Cambridge, UK

## 1 Bregman Clustering

We now show a correspondence between the information gain objective (3) of the main paper and the Bregman clustering objective [1].

In the main paper we specify the partitioning of data between two layers by means of a directed graph assigning subsets to left and right arcs, which in turn are assigned to a child node. Here, the Bregman clustering is run on sets of instances derived from partitioned each parent node instance set into left and right subsets. If we fix these splits we can now solve the problem of assigning these sets $S_{ii}$ to the new child layer using Bregman clustering.

We denote class counts of set $i$ and class $k$ by $S_{i,k}$ and use $i \in j$ and $j(i)$ to denote the assignment of set $i$ to child node $j$ in the next level. The counts are obtained by summation, $S_{j,k} = \sum_{i \in j} S_{i,k}$. We use $\mu_{j,k} = \sum_{i \in j} S_{i,k} / (\sum_k \sum_{i \in j} S_{i,k})$ to denote normalized histograms.

We start with the objective of Algorithm 1 in [1] and derive the equivalence by elementary manipulations as follows.

$$
\begin{aligned}
\sum_j \sum_{i \in j} |S_i|\, D_{\mathrm{KL}}(S_i \| \mu_j) &= \sum_j \sum_{i \in j} |S_i| \sum_k \frac{S_{i,k}}{|S_i|} \log \frac{\frac{S_{i,k}}{|S_i|}}{\mu_{j,k}} \\
&= \sum_i |S_i| \sum_k \frac{S_{i,k}}{|S_i|} \log \frac{\frac{S_{i,k}}{|S_i|}}{\mu_{j(i),k}} \\
&= \sum_i \sum_k S_{i,k} \left[ \underbrace{\log S_{i,k} - \log |S_i|}_{=:C} - \log \mu_{j(i),k} \right] \\
&= \sum_j \sum_{i \in j} \sum_k S_{i,k} \left[ -\log \mu_{j(i),k} \right] + C \\
&= \sum_j \sum_{i \in j} \sum_k S_{i,k} \left[ -\log \frac{\sum_{s \in j} S_{s,k}}{\sum_k \sum_{s \in j} S_{s,k}} \right] + C \\
&= -\sum_j |S_j| \sum_k \frac{S_{j,k}}{|S_j|} \log \frac{S_{j,k}}{|S_j|} + C \\
&= \sum_j |S_j| H(S_j) + C.
\end{aligned}
\tag{1}
$$

Hence, except for an additive constant $C$ that does not depend on the assignment, the Bregman clustering objective using the KL-divergence is equivalent to the minimum information gain loss objective (3) of the main paper.

## 2    DAG Visualization

Fig. 1 shows a visualization of one of the resulting merged DAGs.

## 3    UCI Experiments

Table 1 lists the UCI multiclass classification data sets we used together with the number of classes, the total number of samples available, and the number of feature dimensions. All data sets have been obtained from the libsvm data set collection page.

| DATA SET | CLASSES | SAMPLES | DIMENSIONS |
|---|---|---|---|
| POKER | 9 | 1025010 | 10 |
| COVTYPE | 7 | 581012 | 54 |
| CODRNA | 2 | 331152 | 8 |
| IJCNN1 | 2 | 141691 | 22 |
| SEISMIC | 3 | 98528 | 50 |
| CONNECT4 | 3 | 67557 | 127 |
| W8A | 2 | 64700 | 300 |
| MNIST | 9 | 60000 | 780 |
| SHUTTLE | 7 | 58000 | 9 |
| PROTEIN | 3 | 24387 | 357 |
| LETTER | 26 | 20000 | 16 |
| PENDIGITS | 9 | 10992 | 16 |
| SECTOR | 105 | 9619 | 55197 |
| USPS | 10 | 9298 | 256 |
| GISETTE | 2 | 7000 | 5000 |
| SATIMAGE | 6 | 6435 | 36 |
| DNA | 3 | 3186 | 180 |
| OIL | 3 | 3000 | 12 |
| VOWEL | 10 | 990 | 10 |
| WEBKB | 5 | 877 | 1703 |
| VEHICLE | 4 | 846 | 18 |
| SVMGUIDE4 | 3 | 612 | 10 |
| FACES-OLIVETTI | 40 | 400 | 10304 |
| SVMGUIDE2 | 3 | 391 | 20 |
| SOY | 3 | 307 | 35 |
| GLASS | 6 | 214 | 9 |
| WINE | 3 | 178 | 13 |
| IRIS | 3 | 150 | 4 |

Table 1: UCI data set characteristics.

The experimental setup is as follows. For each data set all instances from the training, validation, and test set, if available, are combined to a large set of instances. We repeat the following procedure five times: randomly permute the instances, and divide them 50/50 into training and testing set. Train of the training set, evaluate the performance on the test set. We report average multiclass accuracy and standard deviation over these five repetitions.

Tree/DAG parameters: each ensemble contains 8 trees or DAGs. The DAG is grown starting from the fifth layer for a maximum of 55 steps. Each DAG layer has at most 1.2 times the number of nodes as the previous layer, or a maximum of 128. The number of feature tests is the same for trees and DAGs and set to 64 dimensions and 20 thresholds per dimension.

Figures 2 to 5 report the average test set accuracy as a function of the total number of nodes in the model for the 28 UCI data sets.

# References

[1] A. Banerjee, S. Merugu, I. S. Dhillon, and J. Ghosh. Clustering with Bregman divergences. *Journal of Machine Learning Research*, 6:1705–1749, Oct. 2005.

Figure 1: **DAG Visualization.** Colors indicate the most likely classes at each node, and saturation indicates purity. The visual layout of the DAG has been optimized slightly to minimize the sum of absolute horizontal edges from each level to the next.

Figure 2: UCI classification results.

Figure 3: UCI classification results.

Figure 4: UCI classification results.

Figure 5: UCI classification results.