[Reviews · NeurIPS 2013]

Submitted by Assigned_Reviewer_4

the paper starts with the observation that rooted DAG and binary decision trees are quite similar. Fixing the number of child notes in a rooted DAG (RDAG) the authors then show in a series of experiments that these RDAG (and ensembles of those) compare favorably to a number of baselines. As this is a somewhat different type of regularization than is typically used in randomized trees this is a worthwhile empirical observation. In my view the key contribution is the experimentation showing indeed that these RDAG might well have practical importance. While any experimentation can be extended the authors have done a good job in my view covering various aspects.

the authors argue in section 2 that binary decision trees and rooted binary decision DAGs share quite some similarity - this is also what is used in the remainder of the paper. In the abstract, title and introduction however the writing suggests that a much more general similarity is used which I find at least confusing if not even annoying - in that sense the authors should tone down their statements at the beginning of their paper as this is not justified by the rest of the paper.

another detail: the abbreviation DAG is quite standard - but nevertheless the meaning of the abbreviation should be given the first of its occurrence in the text (that is the abstract in this case)
Summary: RDAG (rooted directed acyclic graphs) are explored in this paper (and ensembles) thereof. Experiments show that fixing the number of child notes obtains good results w.r.t. other randomized tree classifiers.

Submitted by Assigned_Reviewer_5

Summary of the paper:
This paper revisits the idea of decision DAGs for classification. Unlike a decision tree, a decision DAG is able to merge nodes at each layer, preventing the tree from growing exponentially with depth. This represents an alternative to decision-trees utilizing pruning methods as a means of controlling model size and preventing overfitting. The paper casts learning with this model as an empirical risk minimization problem, where the idea is to learn both the DAG structure along with the split parameters of each node. Two algorithms are presented to learn the structure and parameters in a greedy layer-wise manner using an information-gain based objective. Compared to several baseline approaches using ensembles of fixed-size decision trees, ensembles of decision DAGs seem to provide improved generalization performance for a given model size (as measured by the total number of nodes in the ensemble).

Quality:
This paper is of generally good quality. The related work seems to have been throughly investigated, the model and algorithms make intuitive sense, and the experiments are fairly compelling and do a good job of investigating the effects of varying different design parameters. One aspect that seems to be missing is a comparison of the training and evaluation times of different approaches. This is an important consideration in these models that I think should be addressed.

Clarity:
The paper is quite clear for the most part. The writing is good, the model is well-presented, and the experiments are fairly easy to comprehend. One minor detail that I am not quite clear on is whether the ensemble is trained with bagging in addition to the other randomized elements of the learning algorithm. Also, the authors suddenly refer to energy in section 3.1, is this the same as the empirical risk? Finally, there is a minor grammatical error at the end of the LSearch description: "for considerably less compute."

Originality:
The ideas underlying this paper are fairly well established (decision DAGs, ensembles, empirical risk minimization and information gain). However, the novelty of this paper is in combining these ideas into a cohesive model, and providing intuitive algorithms to learn it.

Significance:
Random forests are a heavy favorite among machine learning practitioners, and therefore any related method that improves upon them without significantly increasing their computational or implementation overhead could have a large impact.
Summary: The authors present ensembles of randomized decision DAGs in order to improve classification under memory constraints, and cast the learning problem in terms of empirical risk minimization. Relatively thorough experiments on several datasets demonstrate the potential of the method.

Submitted by Assigned_Reviewer_6

This paper proposes a modication to the random forests to make the models more memory-efficient. Instead of using trees, it use DAGs. The paper proposes two methods for automatically learning the DAGs.

The overall quality of this paper is clearly below the threshold for NIPS. First of all, the novelty wrt random forests is very limited. The only modification is that this paper uses DAGs instead trees. Although this paper claims that the advantage of DAG is that it is more memory efficent, but this point is not sufficently demonstrated (more on this later).

Second, the proposed technique is very ad-hoc. Although Section 3 starts with an optimization problem in Eq 2-4. The optimization in Sec 3.1 ends up being some ad-hoc local search methods. The description of the algorithms in Sec 3.1 is extremely hand-waving. I doubt anyone can implement the algorithm with the given description. Also, does the algorithm converge at all (even to some local minimum)?

The experimental results did not really demonstrate the benefit of the proposed approach. The main claim of this paper is that it is more memory efficent. However, the experiments in the paper only try to demonstrate that the proposed method generates models with less nodes. But the number of nodes is only one of the factors of model size (you also need to store other information of the model, e.g. the spliting function, the class distribution at leaf nodes, etc). It is not clear to me how the number of nodes will translate to memory efficency. For instance, if the number of nodes only takes a small percetange of the overall model size, perhapse 3000 nodes and 22000 nodes will take rougly the same amount of memory. In addition, memories are getting cheaper everyday. If the proposed method only reduces the memory by a small amount (say a few KB), it is not relevant for real applications.

Summary: This paper seems to be trivial modification of existing techniques. It is unlikely to have any impact and is clearly below the NIPS standard.
Author Feedback

Author rebuttal: We thank all the reviewers for their efforts. We start this rebuttal by reiterating our contributions, and then address specific concerns, especially those from AR6 where there has clearly been some misunderstanding leading to a serious error in his/her review. We kindly ask that AR6 revisits his/her review in light of our clarifications below.

Contributions:
* We highlight that traditional decision trees grow exponentially in memory with depth, and propose fixed-width rooted decision DAGs as a way of avoiding this. (Noted by AR5).
* We propose and compare two “intuitive” (AR5) algorithms that, within each level, jointly optimize an objective function over both the structure of the graph and the split functions.
* We show that not only do the DAGs dramatically reduce memory consumption, but also can improve generalization (Noted by AR5, but missed by AR6). AR4 found the observations “worthwhile” and AR5 “fairly compelling”. AR5 believes the approach “could have a large impact”.

Concerns of AR6:
* Memory efficiency: AR6 has overlooked some basic properties of data structures, leading to what we believe is an error in his/her review (i.e. that the memory footprint of a tree/DAG is somehow not O(N)). To store a tree/DAG we must store: (a) the structure; (b) the split functions; (c) the leaf distributions. For trees, (a,b,c) each require O(N) where N is the number of nodes (e.g. the leaf distributions require O(0.5xN)=O(N)). With a DAG, (a,b) still require O(N) memory, though now (c) requires only O(M) where M << N is the number of leaf nodes. Comparing trees/DAGs, the amount of memory required per split node is the same ((a) trees are typically implemented with child pointers unless they are completely balanced, and (b) the split functions are identical in both cases), and per leaf node is also the same ((c) the distributions are identical in both cases). Therefore, using the number of nodes N as a proxy for actual memory consumption is perfectly valid and our results clearly substantiate our claims. To avoid any future confusion, we will do a better job of explicitly stating this relationship in the final draft.
* To empirically demonstrate the above, we will include the following concrete examples of memory reduction in the final draft: On the Kinect dataset, the forest of 3 trees occupied 80MB vs 9MB for 3 DAGs. On the Faces dataset the forest of 3 trees occupied 7.17MB vs 1.72MB for 3 DAGs. This is much more than “a few KB” and very relevant for real applications. (We further demonstrate improved generalization of DAGs over trees, substantially so in some cases).
* “Memories are getting cheaper everyday”. By this reasoning, practical applications can only go one level deeper every 18 months (Moore’s law, assuming it continues). Our approach allows us to go as deep as we like (within reason), today. (See also comment on generalization above).
* Novelty: We list our novelties above. These include new algorithms that jointly optimize structure and features, and “compelling” (AR5) empirical observations.
* “Ad-hoc” technique: while our optimization algorithms are local, they are “intuitive” (AR5) and (greedily) optimize a well-defined energy function (eq.2). The optimizations are guaranteed to converge (at each iteration the energy cannot increase). The description was necessarily short due to space limitations, but we will try to add more detail to the final revision to improve the clarity here.
* “Unlikely to have any impact”: We believe the dramatic reduction in memory consumption actually makes our approach ideal for real-world memory-constrained applications.

AR4:
* “general similarity [between binary decision trees and rooted decision DAGs … I find at least confusing”: We will happily tone down any unjustified statements. We are however struggling slightly to pin down precisely what the concern is here: is the issue that we are not precise enough with terminology (i.e. we need to be more explicit about rooted decision DAGs as opposed to DAGs in general) and that we thus overclaim in stating that “ensembles of decision DAGs … generalize randomized decision forests”? If there is a way to feed back more detail here (perhaps via the AC’s final report) we would very much welcome this to help us improve the final revision. Regardless, we will do our utmost to ensure the abstract/title/introduction is made more specific and correct.

AR5:
* Training/testing times: As an example, on the Kinect dataset, training 3 trees took 32mins vs 50mins for 3 DAGs. DAGs do take a little longer to train, but can achieve both smaller memory footprint and improved generalization. We will add these timings to the final revision.
* Variants of bagging were used in all the experiments (e.g. the Kinect training images are samples from a generative model).
* Energy vs empirical risk: the entropy-based energy can be viewed as minimizing a log-loss. We will clarify for the final revision.

Minor points:
* We will address all remaining minor suggestions in the final revision.